# Proteome Analysis of Male Accessory Gland Secretions in the Diamondback Moth, *Plutella xylostella* (Lepidoptera: Plutellidae)

**DOI:** 10.3390/insects14020132

**Published:** 2023-01-27

**Authors:** Li-Juan Wu, Fan Li, Yue Song, Zhan-Feng Zhang, Yong-Liang Fan, Tong-Xian Liu

**Affiliations:** 1State Key Laboratory for Crop Stress Biology in Arid Areas, College of Plant Protection, Northwest A&F University, Xianyang 712100, China; 2Key Laboratory of Integrated Pest Management on Crops in Northwestern Loess Plateau, Ministry of Agriculture and Rural Affairs P. R. China, College of Plant Protection, Northwest A&F University, Xianyang 712100, China; 3Institute of Agricultural Sciences of Suqian, Jiangsu Academy of Agricultural Sciences, Suqian 223800, China; 4Institute of Entomology, Guizhou University, Guiyang 550025, China

**Keywords:** *Plutella xylostella*, seminal fluid proteins, male accessory gland, accessory gland proteins, proteomics

## Abstract

**Simple Summary:**

Male accessory gland proteins (ACPs) play an important role in insect reproduction. Identification of ACPs is crucial to study female reproduction and physiology in a given insect species. The diamondback moth, *Plutella xylostella* (L.), is one of the most destructive insect pests, affecting cruciferous vegetables all over the world. In this study, two different proteomic methods were used to investigate the ACPs in *P. xylostella*. In total, 123 putative secreted ACPs were identified. Comparing *P. xylostella* with other four species of insect ACPs, some new insect ACPs were discovered.

**Abstract:**

In insects, male accessory gland proteins (ACPs) are important reproductive proteins secreted by male accessory glands (MAGs) of the internal male reproductive system. During mating, ACPs are transferred along with sperms inside female bodies and have a significant impact on the post-mating physiology changes of the females. Under sexual selection pressures, the ACPs exhibit remarkably rapid and divergent evolution and vary from species to species. The diamondback moth, *Plutella xylostella* (L.) (Lepidoptera: Plutellidae), is a major insect pest of cruciferous vegetables worldwide. Mating has a profound impact on the females’ behavior and physiology in this species. It is still unclear what the ACPs are in this species. In this study, two different proteomic methods were used to identify ACPs in *P. xylostella*. The proteins of MAGs were compared immediately before and after mating by using a tandem mass tags (TMT) quantitative proteomic analysis. The proteomes of copulatory bursas (CB) in mated females shortly after mating were also analyzed by the shotgun LC-MS/MS technique. In total, we identified 123 putative secreted ACPs. Comparing *P. xylostella* with other four insect ACPs, trypsins were the only ACPs detected in all insect species. We also identified some new insect ACPs, including proteins with chitin binding Peritrophin-A domain, PMP-22/ EMP/ MP20/ Claudin tight junction domain-containing protein, netrin-1, type II inositol 1,4,5-trisphosphate 5-phosphatase, two spaetzles, allatostatin-CC, and cuticular protein. This is the first time that ACPs have been identified and analyzed in *P. xylostella*. Our results have provided an important list of putative secreted ACPs, and have set the stage for further exploration of the functions of these putative proteins in *P. xylostella* reproduction.

## 1. Introduction

Insect seminal fluid, the non-sperm component of ejaculation in male insects, is a complex mixture of biomolecules including proteins/peptides, lipids, prostaglandins [1], steroid hormones [2], etc. [3]. Seminal fluid proteins (SFPs) are typically produced in secretory tissues (accessory glands, ejaculatory duct, etc. [3]) of the male reproductive tract, and are transferred together with sperms to the female reproductive tract during copulation. Therefore, their functions extend from males to females. The SFPs play a vital role in insect reproduction, such as in sperm storage, sperm activation [3,4], and modification of female post-mating physiology and behavior, including decreasing receptivity of remating, affecting longevity, forming the mating plug, etc. [3,5,6]. The proteins/peptides secreted from the male accessory gland (MAG) are referred to as accessory gland proteins (ACPs), and they account for the majority of SFPs in most insect species. The first identified and most well-known ACP is the Acp70A or sex peptide (SP). The SP represses female remating by inhibiting sex pheromone biosynthesis and stimulates oviposition by enhancing the juvenile hormone (JH) biosynthesis in *Drosophila melanogaster* [7,8], but the long-term maintenance of the mating effect requires the participation of sperms [9]. A pheromonostatic peptide (PSP), which inhibits female sex pheromone biosynthesis, as well as calling behaviors have been reported in the corn earworm, *Helicoverpa zea* [10], but it is very interesting that the PSP homolog has not yet been detected in any other insect species. 

Identification of ACPs is challenging especially in non-model organisms, though ACPs are essential for insect reproduction and their functions are largely conserved. So far, it has been found that a large proportion of individual ACPs show unusual, rapid evolution at the primary sequence level [3]. The rapid evolution of ACPs is generally believed to be subject to sexual selection forces, such as male–female co-evolutionary conflict and sperm competitiveness [11,12]. To aid in the identification of ACPs, many molecular and genetic methods combined with bioinformatics analysis have been developed and are used widely. Proteomic research on ACPs has been performed in flies, mosquitoes, moths, crickets, honeybees, and beetles [13,14,15,16,17]. By combining EST with proteomic analysis, 51 novel SFPs were identified, and the majority of annotated SFPs in two species of *Heliconius* butterflies were predicted to be chymotrypsins [16]. Proteome analysis of MAG in *Bactrocera dorsalis* identified 90 ACPs, including a novel ACP, a juvenile hormone-binding protein (JHBP) which impact reproductive events in the female [18]. In another study on *Nilaparvata lugens*, 94 putative secreted SFPs were identified by proteome studies using high-throughput mass spectrometry. Selenoprotein, epidermal growth factor (EGF) domain-containing proteins, mesencephalic astrocyte-derived neurotrophic factor, and a neuropeptide ion transport-like peptide were discovered as new insect SFPs [19]. 

The diamondback moth, *Plutella xylostella* L. (Lepidoptera: Plutellidae), is a worldwide major insect pest of cruciferous vegetables, such as cabbage, cauliflower, and many other important economic food crops [20,21,22]. The total global economic cost of its destruction and management is estimated at USD 4–5 billion per year [23]. Due to the extensive use of chemical pesticides, all major kinds of insecticides have been ineffective against *P. xylostella*, making it challenging to control [20]. Studying the ACPs may provide a novel target by disrupting mating to control this notorious pest. Although some data on gene annotation and whole-genome sequencing regarding *P. xylostella* have been described previously [24], there is currently little information available about the ACP genes involved in this species reproduction. In the present study, we performed a comparative proteomic analysis on virgin and mated MAGs of *P. xylostella* using tandem mass tag (TMT) technology. Meanwhile, we performed shotgun mass spectrometry-based proteomic analyses to investigate the profiling of mated copulatory bursa (CB). Combined, we had obtained tissue-specific RNA sequencing (RNA-seq) datasets of the MAGs to identify the transferred ACPs of *P. xylostella*. This work has set the stage for further confirmation of the functions of the putative proteins.

## 2. Materials and Methods

### 2.1. Insects 

The original strain of *P. xylostella* was collected from a cabbage greenhouse in Yangling (34°30′ N, 108°08′ E), Shaanxi, China, and established a stable laboratory population in our laboratory over 2 months. The laboratory population was reared on cabbage (*Brassica oleracea* L. var. Qingan 70) in cages (length, width, and height of 50 cm × 50 cm × 50 cm). The cages placed in artificial climate chamber at 25 ± 1 °C, 60 ± 5% RH, and 16L: 8D. The pupae were collected and maintained individually in 4 mL microtubes with ventilation holes on the surface until the adults emerged. The newly emerged adults were collected and labeled as one-day-old. The adult moths were fed with 10% sucrose solution (Huaguang, Guangdong, China). 

### 2.2. Tissue Collection

During the scotophase, a two-day-old virgin male and a two-day-old virgin female were placed together for copulation. They were placed into a 50 mL microtube, the orifice of which was covered with 80 mesh gauze, and a cotton ball soaked with 10% sucrose solution was placed at the bottom. Copulation was examined every 30 min with a faint red light. A male and female were considered mated if they were in copula for more than 30 min and completed the coupling naturally. Once the male and female naturally finished copulation, they were anesthetized with CO_2_ and immediately dissected in sterile lepidopteran saline [25] on ice under a stereomicroscope (JSZ6, Jiangnan, Nanjing, China). In order to reduce the error, the mated-MAG and mated-CB tissues collected on the same day were divided into 3 equal portions, respectively. The samples were stored at −80 °C immediately. When each collection of samples reached 200 tissues, then further analysis was performed. For each type of sample, three biological replicates were prepared, including mated-MAG and mated-CB. For virgin MAG sample, approximately 200 two-day-old virgin MAGs were collected per sample with three biological replications. Then, three mated-CB samples were mixed as a composite sample for analysis.

### 2.3. Protein Extraction and Digestion 

Protein extraction was conducted using the methods described by Wisniewski et al. (2009) with modifications. A total of 200 μg of proteins and 30 μL of SDT (4% SDS, 100 mM DTT, 100 mM Tris-HCl) lysis buffer were mixed for each sample, then completely homogenized by using a plastic grinding rod and kept it in 100 °C boiling water for 15 min. After centrifuging at 14,000× *g* for 40 min, the supernatant was filtered through 0.22 µm filters. Following that, the BCA Protein Assay Kit (Bio-Rad Corp., Hercules, CA, USA) was used to quantify the filtrate. For each sample, trypsin (Promega Corp., Madison, WI, USA) was used to digest 200 μg of proteins according to the instructions in the filter-aided sample preparation (FASP) [26]. The resulting peptides were collected for subsequent experiments. 

### 2.4. Quantitative Proteomic Analysis of MAGs Based on Tandem Mass Tags (TMT)

Tandem mass tags (TMT) reagent was used to label 100 μg of the resulting peptide mixture of each sample (virgin-MAG and mated-MAG, 3 biological replicates each) in accordance with the manufacturer’s instructions (90064B, Thermo Scientific, Waltham, MA, USA). The Pierce high pH reverse-phase fractionation kit (84868, Thermo Scientific, Waltham, MA, USA) was used to fractionate the labeled peptides into 10 fractions by increasing acetonitrile step-gradient elution. Each fraction was injected for liquid chromatography-mass spectrometry/MS (LC-MS/MS) analysis. LC-MS/MS analysis was conducted for 60 min using a Q Exactive mass spectrometer coupled to an Easy nLC (Thermo Scientific, Waltham, MA, USA). The protocol was carried out according to the methods described in previous studies [27]. 

### 2.5. Qualitative Proteomic Analysis of Mated-CB Based on Shotgun LC-MS/MS

Considering that not all ACPs were secreted via a classical pathway and had a well-defined signal peptide at N-terminal, herein, we used qualitative proteomic analysis of mated-CB to identify as many ACPs as possible. After the protein extracts of mated-CB were digested by trypsin, the obtained peptide mixture of the mated-CB sample was separated into 10 separate fractions in accordance with the manufacturer’s instructions using the Pierce high pH reverse-phase fractionation kit (QL225317A, Thermo Scientific, Waltham, MA, USA). For LC-MS/MS analysis, 5 μL of each fraction was injected. LC-MS/MS analysis was conducted for 60 min using a Q Exactive mass spectrometer coupled to an Easy nLC (Thermo Scientific, Waltham, MA, USA). The protocol was carried out according to the methods described in previous studies [28].

### 2.6. Data Analysis 

The MASCOT engine (Matrix Science, London, UK; version 2.2) was used to search each sample’s MS raw data and embedded into the software Proteome Discoverer 1.4 (Thermo Scientific, Waltham, MA, USA) for quantitative analysis and identification. The related parameters were as follows: peptide mass tolerance was 20 ppm and fragment mass tolerance was 0.1 Da. The proteolytic enzyme was trypsin, with two maximum missed cleavages. The fix modifications were carbamidomethyl (C) and oxidation (M), as variable modifications. Based on a false discovery rate (FDR) of 1% and 99% confidence, the peptides were extracted. The database used in this study was generated from *P. xylostella* MAGs transcriptome (accession number SRP306271 in the Sequence Read Archive (SRA) of NCBI) containing 28066 amino acid sequences. The putative coding sequences (CDSs) of transcriptome were predicted using Transdecoder software (https://github.com/TransDecoder/TransDecoder/releases; accessed on 23 January 2023, version 3.0.0), and the longest CDS of each transcript was taken as the predicted CDS sequence of that transcript. To improve the confidence of prediction, homology was explored by comparison with Uniprot databases (ftp://ftp.uniprot.org/pub/databases/uniprot; Database fetching date: 27 April 2016) and Pfam databases (ftp://ftp.ebi.ac.uk/pub/databases/Pfam; Database fetching date: 27 April 2016), and Evalue cut-off was set at 1 × 10^–5^. For quantitative proteomic analysis of virgin- versus mated-MAGs, statistical analysis were performed according to the methods described in previous studies [29,30,31,32]. A fold change value greater than ± 1.5 and *p* < 0.05 were used as markers to identify the differentially abundant proteins (DAPs). All DAPs were tested for subcellular localization prediction by the online tool WoLF PSORT (https://wolfpsort.hgc.jp/, accessed on 23 January 2023). 

### 2.7. Identification of ACPs of P. xylostella

The proteins with the following standards were considered to be predicted secreted ACPs: (1) The DAPs contained signal peptides. All coding sequences (CDS) of DAPs were tested for signal peptide prediction by the online tool SignalP 5.0 (https://services.healthtech.dtu.dk/service.php?SignalP-5.0, accessed on 23 January 2023). To improve the predicted signal peptide detection, the proteins without signal peptides were re-predicted CDS sequences from unigene nucleotide sequences by the online tool ESTScan (https://myhits.sib.swiss/cgi-bin/estscan, accessed on 23 January 2023), and signal peptide detection was performed with the new predicted CDS sequences. (2) The DAPs did not possess a signal peptide, but were detected in mated-CB. (3) In addition, the DAPs that did not satisfy the above two criteria were considered as unconfirmed ACPs.

### 2.8. Annotation of ACPs and Comparison with Other Insects

We used the annotation and classification methods described by Yu et al. (2015) for identifying ACPs. Briefly, in addition to machine annotation, the ACPs were manually annotated by combining NCBI blast results, GO terms, and conserved domains. Combining the descriptions of conserved domains from NCBI, InterProScan (http://www.ebi.ac.uk/interpro/search/sequence/, accessed on 23 January 2023) and SMART (http://smart.embl-heidelberg.de/, accessed on 23 January 2023), the ACPs were categorized into one of the following groups: “cell and extracellular structure”, “metabolism”, “proteolysis regulators” (including proteases and protease inhibitors), “RNA and protein synthesis”, “signal transduction”, “protein modification machinery”, “transporters and protein export machinery”, and “other” (including immune, chitin binding proteins, oxidoreductase, chaperone, ubiquitination pathway, reverse transcript, apoptosis, and cell cycle control). In this study, proteins having no known function were categorized as “unknown.”

*D. melanogaster*, *Aedes aegypti*, *Apis mellifera*, *Heliconius erato*, and *H. melpomene* seminal fluid proteome sequences, together with *Homo sapiens*, were chosen for comparison with *P. xylostella* ACPs (Appendix A). We chose these species because they have different reproductive strategies, and they were available to us. Among them, *Heliconius* butterflies and *P. xylostella* belong to the group of lepidopteran insects. *D. melanogaster* has the most studied ACPs in all insect species [3]. The mosquito is similar to *P. xylostella*, in that mating can cause females to be resistant to subsequent copulations and can enhance egg production [14,33,34]. *A. mellifera* represents a social insect species. *H. sapiens* represents mammals, and was selected to serve an outgroup control. Mammalian reproduction is completely different from that of insect species. *H. Sapiens* SFPs sequences were obtained from the UniProt database (https://www.uniprot.org/id-mapping; Database fetching date: 13 October 2021) utilizing the ID number given in the reference [35]. *D. melanogaster* SFPs sequences were obtained from the Flybase database (http://flybase.org/download/sequence/batch#; Database fetching date: 13 October 2021) utilizing the ID number given in the reference [12]. *A. mellifera* SFPs sequences were obtained from NCBI (https://www.ncbi.nlm.nih.gov/sites/batchentrez; Database fetching date: 13 October 2021) utilizing the ID number given in the reference [13]. *A. aegypti* SFPs sequences were obtained from the VectorBase database (https://vectorbase.org/vectorbase/app/; Database fetching date: 13 October 2021) utilizing the ID number given in the reference [36]. SFPs sequences of *Heliconius* butterflies were obtained from NCBI (https://www.ncbi.nlm.nih.gov/sites/batchentrez; Database fetching date: 13 October 2021) utilizing the ID number given in the reference [16]. The ID numbers and amino acid sequences of SFPs for these species can also be obtained in the Appendix A. The signal peptide prediction of these SFPs were performed as mentioned in the identification of ACPs in *P. xylostella*. The conserved domain predictions of these SFPs were performed by the NCBI online tool Batch Web CD-Search (http://www.ncbi.nlm.nih.gov/Structure/cdd/wrpsb.cgi, accessed on 23 January 2023). The *P. xylostella* ACPs that shared a conserved domain with SFPs from the other four insects were marked as “Domain”, and the rest of the proteins were analyzed based on local blastp by TBtools software [37]; blastp hits (Evalue < 10^−5^) with SFPs of the other four insect were marked as “Blast”. A comparison of SFPs among insect species was performed using a similar method.

### 2.9. Phylogenetic Analysis

We used ClustalW to align the trypsin (serine protease) conserved domains of the ACPs of *P. xylostella* with the serine proteases from the other four insect species (Appendix A). The maximal likelihood (ML) and Jones–Taylor–Thornton (JTT) substitution models were used to construct the phylogenetic tree using Mega software (version 5.05), followed by a 1000-replication bootstrap test to analyze homologous relationships.

### 2.10. Tissue-Specific Expression Analysis by qRT-PCR

The testis (TE), MAG, and vas deferen (VD) of the male reproductive system of *P. xylostella* (Figure 1) were dissected from two-day-old virgin males. The CB was dissected from two-day-old virgin females. All of the tissues, dissected from approximately 50 individuals, were pooled into a biological sample. Three replicates were prepared for each tissue. The total RNA was isolated using RNAiso Plus (Takara, Dalian, China) based on the instructions provided by the manufacturer. The reverse transcription was carried out by a PrimeScript^®^ RT Reagent Kit with gDNA Eraser (Takara, Dalian, China) in a reaction mixture of 10 µL, with 900 ng of total RNA. qRT-PCR was carried out in accordance with the method described by Wei et al. (2015) [18]. Statistical analyses were carried out utilizing the IBM SPSS Statistics package 19 (SPSS Inc., Chicago, IL, USA). The different tissues were compared using one-way analysis of variance, followed by Duncan’s test, at a significance level of *p* < 0.05. The data are displayed as the means ± standard error (SE). The primers that were used for tissue-specific expression of ACP genes using qRT-PCR are given in Appendix A. The housekeeping ribosomal protein S13 gene (RPS13) was used for data normalization. 

## 3. Results

### 3.1. Identification of DAPs between Mated and Virgin MAG

A total of 5101 proteins were identified and quantified in at least two of the three biological replicates in two experimental groups between the mated MAG group and the virgin MAG group. The Student’s *t*-test detected statistically significant proteins (*p* < 0.05), and the abundances of proteins that changed by more than 1.5- or less than 0.67-fold were maintained. A total of 197 differentially abundant proteins (DAPs) were detected (Appendix A). The identified DAPs are shown by a volcano plot (Figure 2). We selected the longest CDS to represent the protein containing CDSs of different lengths. Based on this criterion, a total of 168 DAPs were selected, of which 165 proteins were significantly down-regulated and 3 proteins were significantly up-regulated (Appendix A). The signal peptide prediction tool SignalP was used to detect secretory signals for 74 proteins in DAPs, all of which were down-regulated proteins. None of the three up-regulated proteins contained signal peptides.

### 3.2. Identification of Proteins in Mated-CB

Using shotgun proteomics, 20,153 peptides corresponding to 3770 proteins from mated-CB were identified. Of these, there were 115 proteins in common between DAPs and mated-CB. Among them, signal peptides were predicted to be found in 66 proteins, and 53 proteins were exclusively identified in DAPs, 8 of which were predicted to be secreted proteins due to the presence of signal peptides. Two up-regulated DAPs were not detected in mated-CB. Only one up-regulated DAP (Unigene056697_03, predicted as aromatic-L-amino-acid decarboxylase) was detected in mated-CB, but no signal peptide was found. 

### 3.3. Secreted ACPs Identification in the MAGs

In total, 123 ACPs were predicted to be secreted ACPs, and 74 out of the 123 secreted ACPs had signal peptides. A total of 49 out of the 123 secreted ACPs were detected in the mated-CB, while SignalP predicted no signal peptides (Appendix A and Table 1). The remaining 45 proteins of DAPs were predicted to be unconfirmed ACPs, since these proteins neither had a signal peptide nor were detected in the mated-CB.

### 3.4. Subcellular Localization Analysis of ACPs

Combined with results of subcellular location analysis, we discovered that there were significant differences between the predicted secreted ACPs and unconfirmed ACPs (Appendix A and Figure 3). “Extracellular” accounts for the largest percentage of secreted ACPs (53, 33.8%), followed by “nuclear” (31, 19.7%), “cytoplasmic” (28, 17.8%), “mitochondrial” (24, 15.3%), “plasmamembrane” (17, 10.8%), and “other”, which included “lysosomal” (3, 1.9%) and “endoplasmic reticulum” (1, 0.64%). For unconfirmed ACPs, “nuclear” represented the largest percentage (31, 41.9%), followed by “plasmamembrane” (22, 29.7%), “cytoplasmic” (15, 20.3%) and “other”, which included “mitochondrial” (3, 4.1%), “extracellular” (2, 2.7%), and “peroxisomal” (1, 1.4%). 

### 3.5. Functional Classification Analysis of ACPs

According to the sequence annotation, 168 ACPs were categorized into different functional categories. The number of proteins in each functional category was analyzed (Appendix A and Figure 4). The “proteolysis regulators” were the largest catalog of secreted ACPs (38, 30.9%). The next highest were “unknown” (20, 16.3%) and “metabolism” (20, 16.3%), including carbohydrate, lipid, amino acid, and nucleotide. For unconfirmed ACPs, “other” represented the largest percentage (14, 31.1%), followed by “RNA and protein synthesis” (11, 24.4%). 

### 3.6. Comparison of the ACPs of P. xylostella with Other Insects

A total of 95 out of the 168 DAPs had homologues with SFPs in four other insect species as well as *H. sapiens* (Appendix A). It was found that 51, 62, 16, 17, and 7 homologous proteins were discovered in *D. melanogaster*, *A. aegypti*, *A. mellifera*, *Heliconius* butterflies, and *H. sapiens*, respectively. Among the 95 *P. xylostella* homologous proteins, only trypsins were detected in all insect species. According to the results of the analysis, there were a much higher number of trypsins (18) in *P. xylostella* ACPs than in other four insects (*D. melanogaster*, 14; *A. aegypti*, 6; *A. mellifera*, 1; and *Heliconius* butterflies, 15). Interestingly, these trypsins were all predicted to be secreted proteins. Phylogenetic tree analysis showed that 10 of 18 accessory gland protein trypsins were on the same branch in *P. xylostella*. One trypsin showed a closer relationship with *D. melanogaster*; the other one was closer to *A. mellifera;* and the remaining six of eight trypsins were closer to *Heliconius* butterflies (Figure 5). Considering that the *Heliconius* butterflies and *P. xylostella* are lepidopteran insects, the seminal fluid trypsins displayed close relationships with those of non-lepidopteran insects. These results also indicate that seminal fluid proteins may play a role in species-specific reproductive barriers. 

Considering that not all secreted ACPs may be transferred from males to females during mating, we analyzed the proteins that contained signal peptides, but were not detected from mated-CB samples. Eight proteins met the above criteria, and their information, including nucleic acid sequences and the corresponding amino acid sequences, are listed in Appendix A. 

### 3.7. Tissue-Specific Expression Analysis by qRT-PCR

To confirm the proteomic results, we further determined the transcriptional expression profiles of 35 genes in virgin males’ TE, VD, and MAG, as well as the virgin females’ CB, through qRT-PCR. In total, 33 genes showed strong expression in the MAG of *P. xylostella* (Figure 6). Of these 33 genes, 7 trypsins were expressed at levels tens to hundreds of thousands of times greater in the MAG than in the CB tissue. Two exceptional genes were observed: one was trehalase (Unigene 036627_01), which showed a high expression in the VD, and the other was a ecdysteroid-regulated 16 kDa protein (Unigene 012392_01), which showed a high expression in the TE. The gene expression profiles from qRT-PCR were corroborated with our proteomic data.

## 4. Discussion

In insect species, ACPs are synthesized in the accessory gland of males and transferred to females during mating, and they trigger multiple physiological and post-mating behavioral changes [3,19]. Previous studies identifying ACPs were carried out with samples obtained from the female reproductive system after mating [14,17,19]. Therefore, some ACPs that are low in abundance due to dilution in the female-derived proteins or rapid processing in females may be missed [12,38]. Our study provides a proteomic-scale view of the secreted ACPs produced by the MAGs in *P. xylostella*. We performed TMT quantitative proteomics to identify male accessory glands produced ACPs in *P. xylostella* by comparing changes in the abundance of ACPs immediately before and after mating. In total, 168 ACPs were identified. Among those ACPs, 165 ACPs were significantly down-regulated and 3 ACPs were significantly up-regulated. Most of the ACPs were down-regulated proteins, which also demonstrated that ACPs are transferred to females during mating as well as that the ACP identification method used in this study was efficient. A total of 74 proteins out of 168 ACPs had signal peptides. In order to identify as many ACPs as possible, we used the shotgun qualitative proteomic approach to analyze the proteins of copulatory bursas in mated females. Our data showed that 49 out of 168 ACPs were detected in mated-CB samples, although they were not predicted to contain signal peptides. It was hypothesized that these proteins would be secreted from MAG. Finally, a total of 123 proteins were predicted to be secreted ACPs in *P. xylostella*. 

The 123 secreted ACPs were categorized into 9 groups based on their functions of conserved domains. Most of these proteins, including proteases, protease inhibitors, transporter proteins, and antibacterial proteins, had previously been identified as ACPs in other insects [10,16,18,39]. A total of 45 proteins out of 168 ACPs were predicted to be unconfirmed ACPs, since these proteins neither had a signal peptide nor were detected in the mated sample. Due to technological restrictions, these proteins could not be confirmed to be “true” secreted ACPs in *P.* xylostella (they were either low-abundant or had been processed by the female prior to sampling).

Typically, a signal peptide is used to determine whether a protein is secreted or not via a classical pathway. In this study, 74 out of 123 secreted ACPs had a signal peptide in *P. xylostella*. This is different from what has been found in *D. melanogaster*, where 163 out of 173 SFPs had signal peptides. In *D. melanogaster*, the high percentage of SFPs containing signal peptides could be due to the high quality of protein databases. However, in *Heliconius* butterflies, 44 out of 51 detected SFPs had signal peptides. The high percentage of SFPs containing signal peptides in *Heliconius* butterflies may due to the fact that most of the annotated SFPs were predicted to be chymotrypsins [16]. In *A. aegypti*, 71 out of 93 detected SFPs had signal peptides, as did 21 out of 53 detected SFPs in *A. mellifera*. It is also possible that the lack of signal peptides in secreted ACPs is due to the fact that they are secreted through “apocrine secretion”, whereas cells in the posterior portion of the gland are considered to secrete proteins via granules and/or cell membrane rupture [14,19]. Alternatively, the database contains peptide sequences that lack an N-terminus.

In *P. xylostella,* 38 (27 proteases and 11 protease inhibitors) out of 123 secreted ACPs identified were predicted regulators of proteolysis. Similar results were found in the other four insects, e.g., 21 proteases and 14 protease inhibitors in *D. melanogaster*; 20 proteases and 1 protease inhibitor in *A. aegypti*; 5 proteases in *A. mellifera;* and 15 proteases and 2 protease inhibitors in *Heliconius* butterflies. Proteolysis must be strictly regulated to avoid premature activation of pathways or tissue injury; therefore, ACPs are commonly rich in proteases and protease inhibitors [3,40,41,42]. Proteolysis regulators are predicted or known to function in seminal fluid in several ways, including sperm storage and competition, as well as sperm protection [43], mediation of females’ ovulation and egg-laying after mating, participation in sperm–egg interactions in collaboration with particular proteases, and activation or inactivation of other reproductive proteins [44,45]. 

In this study, 18 trypsins (47% of proteolysis regulators) were identified in *P. xylostella*. Trypsins were detected in the other four insect species besides *P. xylostella*. The number of trypsins identified in the secreted ACPs of *P. xylostella* was the highest among the 5 species, including 14 trypsins (40% of proteolysis regulators) in *D. melanogaster*, 6 trypsins (29% of proteolysis regulators) in *A. aegypti*, 1 trypsin (20% of proteolysis regulators) in *A. mellifera,* and 15 trypsins (88% of proteolysis regulators) in *Heliconius*. A trypsin-like serine protease is necessary for the full induction of egg-laying after mating in crickets [46]. Disruption of serine protease 2 in *Bombyx mori* and *P. xylostella* resulted in male sterility, which could have been caused by the sperm did not successfully enter the egg [47].

Two apolipoproteins (a apolipophorin-III and a apolipoprotein D) were identified in the secreted ACPs of *P. xylostella*. Apolipophorin-III was not detected in ACPs of other four insect species. Apolipoproteins are proteins that transport hydrophobic substances, such as steroid hormones, lipids, and others, and may be related to immunity [48]. Previous studies have reported that the MAGs of insects can synthesize JH and transfer JH from the male to female reproductive tract during copulation in the long-horned beetle species *Apriona germari* and *Spodoptera litura* [49,50]. Apolipoprotein D serves as a lipocalin and can bind arachidonic acid, a precursor of prostaglandin, specifically for the synthesis of prostaglandin [51,52]. In addition to its role in lipid transportation, apolipophorin III (apoLp-III), a prototypical exchangeable apolipoprotein, is also implicated in innate immunological responses and is present in many insect species [53]. 

A chemosensory protein contains an OS-D domain, also called an odorant binding protein (OBP), which was identified as secreting ACPs in *P. xylostella*. OBPs were thought to be present in the olfactory organs and played roles in olfactory recognition [54,55]. It has been reported that OBPs were detected in the MAGs of some insects, including *D. melanogaster*, *B. dorsalis*, and *N. lugens* [18,19,39]. In *D. melanogaster*, some OBPs showed MAG-specific expression [56]. *Obp10* and *Obp22* can significantly impair female and male reproductive capacity, including oviposition, fecundity, fertility, and the development of spermatozoa in *A. aegypti* [57]. Previous studies found that Or10a, an OBPs receptor protein, was significantly up-regulated in female *D. melanogaster* reproductive tracts after mating [58]. In *Tribolium castaneum*, RNAi of OBPs leads to a decrease in females’ egg production after mating [17]. It is believed that the chemosensory protein is transferred from males to females during copulation, and interacts with the receptors in the female reproductive tract to regulate the female’s reproductive physiology in *P. xylostella*. It may transport pheromones, odorants, or other small molecules to receptors. The exact function of OBP, which was found in ACPs of *P. xylostella*, needs to be investigated in the future. 

Previous studies have demonstrated that males transfer antibacterial proteins to females from MAGs and ED during copulation to improve reproductive success in *D. melanogaster* [18,59]. In this study, two types of lectin, namely C-type and S-type, were identified as secreted ACPs in *P. xylostella*. Galectin (also known as S-type lectin) was detected in *P. xylostella,* but was not found in the SFPs of the other four insects, including *D. melanogaster*, *A. aegypti*, *A. mellifera,* and *Heliconius*. Galectin is able to recognize the conserved amino acid motif of β-galactoside enzyme from pathogenic microorganism and induce immune responses against foreign microorganisms in insects [60]. Studies have revealed that C-type lectin expression will be significantly up-regulated in response to Gram-negative bacterial infection in *D. melanogaster* [61]. In *D. melanogaster*, C-type lectins are required in the SP pathway [62]. Two cysteine-rich secretory proteins (CRISPs) were also identified as secreted ACPs in *P. xylostella*. It has previously been reported that the CRISPs gene has a high level of expression in MAGs and EDs in *B. dorsalis*, and may possibly serve an immune-related function in reproductive tissues [63]. CRISP, which is involved in spermatogenesis, was also detected in mammalian testes [64]. 

Comparing *P. xylostella* with the other four insect ACPs, we identified some new insects ACPs in this proteome study, including proteins with chitin binding Peritrophin-A domain, PMP-22/EMP/MP20/Claudin tight junction domain-containing protein, netrin-1, type II inositol 1,4,5-trisphosphate 5-phosphatase, two spaetzles, allatostatin-CC, and cuticular protein. In particular, peritrophic matrix proteins of insects and animal chitinases have the Peritrophin-A domain, which is present in chitin-binding proteins. A protein with chitin binding Peritrophin-A domain is a secreted ACP in *P. xylostella,* and may have a role in reproduction. The PMP-22/EMP/MP20/Claudin tight junction domain-containing protein is a member of the claudins family, and performs diverse functions at specialized cell–cell contact [65]. In human, claudins are related to fertility [66]. The disorganization of claudin-11 expression in Sertoli cells could be one of the factors affecting spermatogenesis in infertile men [67]. Netrin is a diffusible laminin-like protein. In *D. melanogaster*, netrin can affect the fertility of both females and males. Lacking the netrin gene, males significantly reduce the egg-laying capability of wild-type females, and females produce fewer fertilized eggs than wild-types [68]. In this study, eight secreted ACPs were predicted to contain signal peptides, but were not identified in mated-CB, including two spaetzles, allatostatin-CC, and cuticular protein. Spaetzle, a ligand for Toll, performs a significant role in innate immunity by activating the Toll signaling pathway to promote the production of antimicrobial peptides [69,70]. Allatostatin-C was originally identified in *Manduca sexta*, and can significantly inhibit juvenile hormone biosynthesis in vitro by the corpora allata [71]. Allatostatin-CC, as an allatostatin-C paralog, has a non-amidated C-terminal end [72,73]. In vitro, allatostatin-C can inhibit activity of the MAGs and gonoducts in the silkworm *B. mori*, and plays a role in regulation of seminal fluid movement during copulation [74]. Cuticular proteins serve as major components of the insect cuticle [75]. It has been reported that cuticle proteins can enhance eggshells during early embryogenesis in *N. lugens* [76]. We speculate that these eight proteins either did not transfer to the female during mating and only play roles in the male, or that they had been processed by the female reproductive tract prior to sampling. These proteins are worthy of further exploration. Another new ACP, Type II inositol 1,4,5-trisphosphate 5-phosphatase (Inpp5b), was identified in this study. To the best of our knowledge, Inpp5b was identified as an ACP for the first time in insects. Type II inositol polyphosphate 5-phosphatase (Inpp5b) is widely expressed. In mice, if the Inpp5b gene is disrupted, sperm motility and the ability to fertilize eggs can be reduced [77]. Inpp5b may play a similar role in *P. xylostella* as it does in mice.

## 5. Conclusions

In this study, we identified 123 putative secreted ACPs of *P. xylostella* using two different proteomic methods. A total of 20 out of 123 secreted ACPs were functionally unknown proteins. For the other 103 secreted ACPs, we were able to categorize them, according to their functions, into 8 groups. This included regulators of proteolysis, signal transduction, transporters, and protein export machinery, and immunity was the most important. Our results provide a foundation for future functional studies to explore these ACPs in female behavior and reproduction in *P. xylostella*. In practice, the new knowledge of insect-specific ACPs could aid in the discovery of novel pesticide target sites for pest population control in *P. xylostella*. Meanwhile, the present study provides an important platform for research regarding ACPs in *P. xylostella*.

## Figures and Tables

**Figure 1 insects-14-00132-f001:**
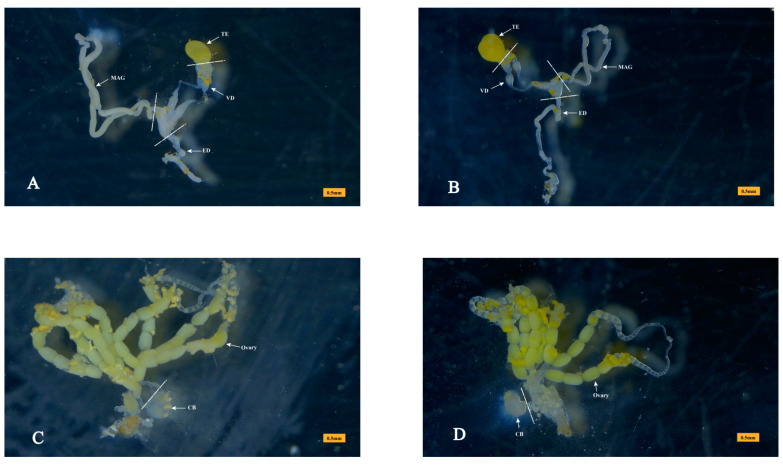
Tissues dissected for sample preparation. (**A**) The reproductive system of virgin male *Plutella xylostella*. Approximately 200 male accessory glands (MAGs) were dissected as a virgin-MAG protein sample, with 3 biological replicates. For qRT-PCR, testis (TE), MAG, and vas deferen (VD) were dissected from 50 virgin males, and 3 biological replicates were performed for each tissue. Ejaculatory duct (ED) is also shown. (**B**) The reproductive system of mated male *P. xylostella*. A mated-MAG protein sample was dissected from approximately 200 mated males, with 3 biological replicates. (**C**) The reproductive system of a virgin female *P. xylostella*. The copulatory bursa (CB) was dissected from 50 virgin females for qRT-PCR, and 3 biological replicates were performed. (**D**) The reproductive system of mated female *P. xylostella*. Approximately 200 CB were dissected as a mated-CB protein sample with 3 biological replicates. Each tissue sample was dissected from the dotted line.

**Figure 2 insects-14-00132-f002:**
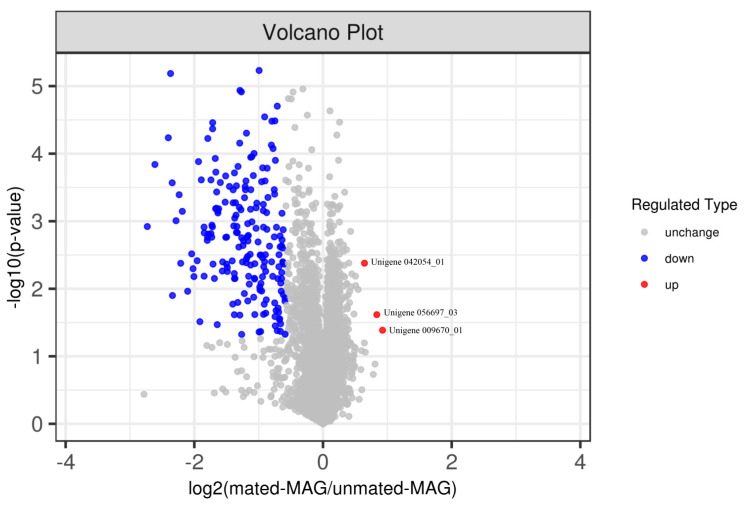
Volcano plot of the differentially abundant proteins (DAPs). The x-axis represents a log2 fold change (mated-male accessory gland (MAG)/unmated-MAG) and the y-axis represents a negative logarithm (log10 scale) of the *p*-value. Each dot represents a protein. Red dots indicate up-regulated proteins. Blue dots indicate down-regulated proteins. Grey dots represent proteins with non-significant differences (from Student’s *t*-test). The farther away from zero that the point is, the more significant the difference.

**Figure 3 insects-14-00132-f003:**
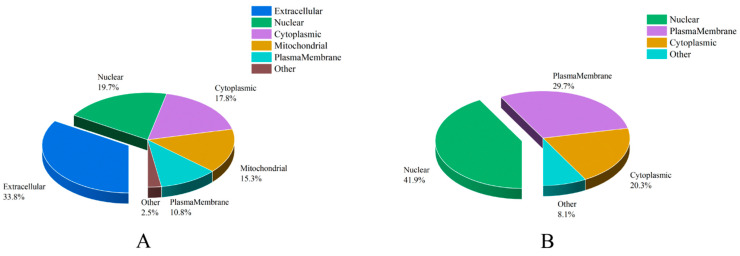
The distribution of subcellular localization of accessory gland proteins (ACPs) in *Plutella xylostella*. (**A**) the distribution of subcellular localization of the predicted secreted ACPs. (**B**) the distribution of subcellular localization of the unconfirmed ACPs.

**Figure 4 insects-14-00132-f004:**
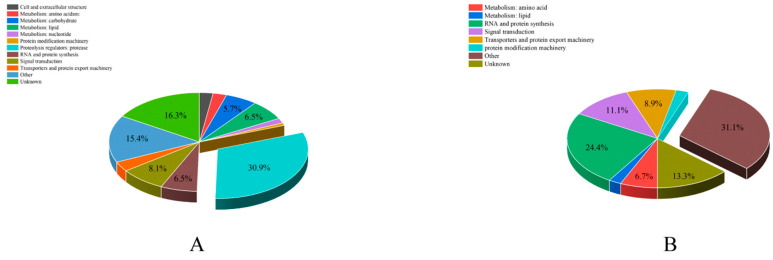
The distribution of functional categories of accessory gland proteins (ACPs) in *Plutella xylostella*. (**A**) functional category analysis of the predicted secreted ACPs. (**B**) functional category analysis of the unconfirmed ACPs.

**Figure 5 insects-14-00132-f005:**
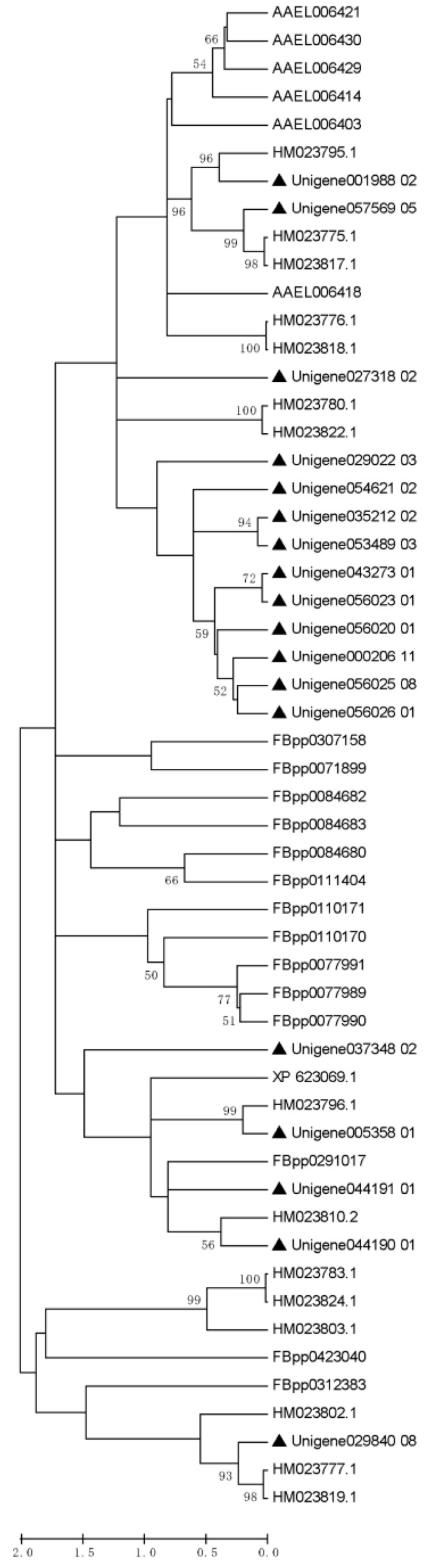
The phylogenetic analysis of seminal fluid trypsins among different insect species. The phylogenetic tree was constructed by the maximum likelihood method and the Jones–Taylor–Thornton (JTT) substitution model using deduced amino acid sequences of the conserved domains of seminal fluid trypsin. The phylogeny was tested with 1000 bootstrap replications using the bootstrap method. Sequences beginning with “Unigene” represent transcriptome unigene IDs of *Plutella xylostella* seminal fluid trypsins, indicated by the black triangle. Sequences beginning with “FBpp” represent IDs of *Drosophila melanogaster* seminal fluid trypsins (http://flybase.org/, accessed on 23 January 2023). Sequences beginning with “AAEL” represent IDs of *Aedes aegypti* seminal fluid trypsins (http://www.vectorbase.org/, accessed on 23 January 2023). Sequences beginning with “XP” represent IDs of *Apis mellifera* seminal fluid trypsins (http://www.ncbi.nlm.nih.gov/, accessed on 23 January 2023). Sequences beginning with “HM” represent IDs of *Heliconius erato* and *H. melpomene* seminal fluid trypsins (http://www.ncbi.nlm.nih.gov/, accessed on 23 January 2023).

**Figure 6 insects-14-00132-f006:**
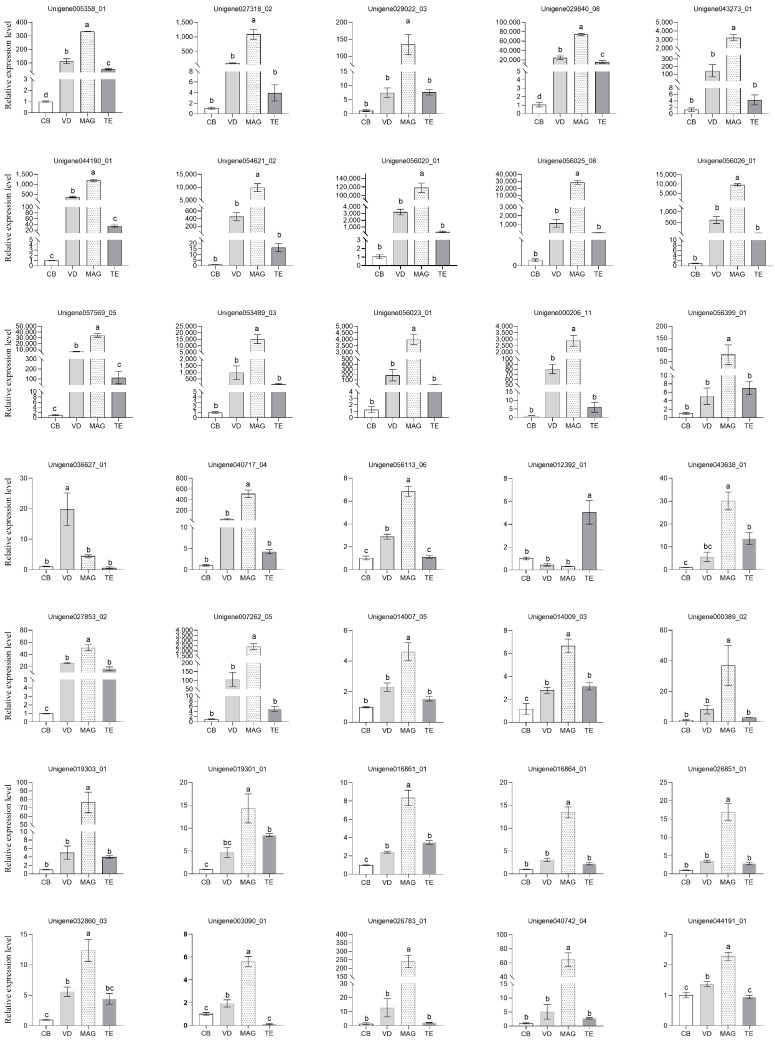
Analysis of the gene expression profiles of accessory gland proteins by qRT-PCR. The X-axis shows the different tissues, including testis (TE), male accessory gland (MAG), vas deferen (VD), and female copulatory bursa (CB). The Y-axis shows the relative mRNA expression level. All data are expressed as the means ± standard error (SE). Relative expression levels were calculated based on the value in CB tissue, which was ascribed an arbitrary value of 1. Different letters above the bars indicate significant differences based on Duncan’s test at a significance level *p* < 0.05.

**Table 1 insects-14-00132-t001:** Identified secreted ACPs of *Plutella xylostella*.

Gene Name	Functional Category	*D. melanogaster*	*A. aegypti*	*A. mellifera*	*Heliconius* Butterflies
Cell and extracellular structure					
Collagen (2)	Cell and extracellular structure		T		
Tetraspanin	Cell and extracellular structure		T		
Metabolism					
Gamma-glutamylcyclotransferase	Metabolism: amino acid				
Aromatic-L-amino-acid decarboxylase	Metabolism: amino acid		T		
Vanin-like protein	Metabolism: amino acid				
Beta-galactosidase	Metabolism: carbohydrate		T	T	
Beta-mannosidase	Metabolism: carbohydrate	T	T		
C-type lysozyme	Metabolism: carbohydrate				
Chitinase	Metabolism: carbohydrate	T	T	T	
Trehalase (2)	Metabolism: carbohydrate	T			
Hyaluronidase	Metabolism: carbohydrate				
Lipase (2)	Metabolism: lipid	T	T		
Phospholipase	Metabolism: lipid	T	T		
Glycosylphosphatidylinositol-specific phospholipase C	Metabolism: lipid	T			
Palmitoyl-protein thioesterase	Metabolism: lipid				
Lysophospholipid acyltransferase	Metabolism: lipid	T			
Phospholipid phosphatase	Metabolism: lipid				
Venom acid phosphatase	Metabolism: lipid		T	T	
Adenosine deaminase (2)	Metabolism: nucleotide	T	T		
Other					
Protein with chitin binding Peritrophin-A domain	Other: chitin binding				
Cysteine-rich secretory protein (2)	Other: immune	T	T	T	
C-type lectin	Other: immune	T	T		
Galectin	Other: immune				
Von Willebrand factor	Other: immune		T		
Immune-related protein	Other: immune				
Gag-pol polyprotein precursor	Other: reverse transcript				
Reverse transcriptase	Other: reverse transcript				
Aldo-keto reductase (2)	Other: oxidoreductase				T
Peptidylglycine alpha-hydroxylating monooxygenase (2)	Other: oxidoreductase	T			
Gamma interferon inducible Lysosomal thiol reductase	Other: oxidoreductase		T	T	
Senecionine N-oxygenase	Other: oxidoreductase			T	
Sulfhydryl oxidase	Other: oxidoreductase	T	T		
GMC oxidoreductase	Other: oxidoreductase	T	T	T	
DnaJ-class molecular chaperone	Other: chaperone		T		
Cuticle protein	Other: protection				
Protein modification machinery					
Glutaminyl-peptide cyclotransferase	Protein modification machinery		T		
Proteolysis regulators					
Lysosomal aspartic protease	Proteolysis regulators: protease		T		
Furin-like protease	Proteolysis regulators: protease		T		
Serine carboxypeptidase (3)	Proteolysis regulators: protease			T	
Zinc carboxypeptidase (4)	Proteolysis regulators: protease	T	T		
Trypsin (18)	Proteolysis regulators: protease	T	T	T	T
Serpin (7)	Proteolysis regulators: protease inhibitor	T	T		T
Cysteine proteinase inhibitor (4)	Proteolysis regulators: protease inhibitor	T	T		
RNA and protein synthesis					
Endonuclease	RNA and protein synthesis		T		
Heterogeneous nuclear ribonucleoprotein	RNA and protein synthesis				
Endoribonuclease (2)	RNA and protein synthesis				
RNA polymerase II transcription elongation factor	RNA and protein synthesis				
Polypyrimidine tract-binding protein	RNA and protein synthesis				
Transcription activator	RNA and protein synthesis				
Translation initiation factor 4E-binding protein	RNA and protein synthesis				
Signal transduction					
Ecdysteroid-regulated 16 kDa protein (2)	Signal transduction	T	T	T	
Spaetzle (2)	Signal transduction				
Nucleobindin	Signal transduction	T	T		
Phosphatidylethanolamine-binding protein	Signal transduction	T	T	T	
Chemosensory protein	Signal transduction			T	
PMP-22/EMP/MP20/Claudin tight junction domain-containing protein	Signal transduction				
Netrin-1	Signal transduction				
Allatostatin-CC	Signal transduction				
Transporters and protein export machinery					
Type II inositol 1,4,5-trisphosphate 5-phosphatase	Transporters and protein export machinery				
ADP-ribosylation factor-like protein	Transporters and protein export machinery		T		
Apolipophorin-III	Transporters and protein export machinery				
Apolipoprotein D	Transporters and protein export machinery	T	T		
Unknown					
Hypothetical protein (blast) (3)	Unknown	T	T		
Unknown (17)	Unknown				

Numbers following gene names indicate the number of proteins detected. The letters “T” indicate that the sequences share a conserved domain or that they exhibit blastp results (Evalue < 10^−5^) with the other four insect SFPs. Analysis of the conserved domains of insect ACPs was conducted using the NCBI online tool Batch Web CD-search (http://www.ncbi.nlm.nih.gov/Structure/bwrpsb/bwrpsb.cgi, accessed on 23 January 2023). A local blastp by TBtools software was conducted based on the similarity criterion of an e-value < 10^−5^ to compare *Plutella xylostella* ACPs with other four insect SFPs. *D. melanogaster*, *Drosophila melanogaster*; *A. aegypti*, *Aedes aegypti*; *A. mellifera*, *Apis mellifera*; *Heliconius* butterflies, *Heliconius erato* and *H. melpomene*.

## Data Availability

The TMT proteomics mass spectrometry data are available via ProteomeXchange with identifier PXD038931. The shotgun proteomics mass spectrometry data are available via ProteomeXchange with identifier PXD039074. The rest of the data are contained within the article or Appendix A.

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
