# Peer review of "Proteome Analysis of Male Accessory Gland Secretions in the Diamondback Moth, Plutella xylostella (Lepidoptera: Plutellidae)"

_insects, 2023, doi:10.3390/insects14020132_

Round 1

Reviewer 1 Report

MS ID: 2109269 Title: Proteome analysis of male accessory gland secretions in the diamondback moth, Plutella xylostella (Lepidoptera: Plutellidae)

General comments:

The authors performed an experiment on the proteome analysis in male reproductive organ of P.. xylostella. The topics are interesting. In general, the manuscript is well written and the main results are correctly interpreted. However, I missed some of the information regarding the data analysis that  needs to be addressed properly.  

However, the manuscript may be accepted after making a minor revision.

Specific comments:

In section 2.6. Data Analysis: I do not know, where the authors used the tukey's test since the they mentioned the Duncan’s test (2.6).

In result section 3. Needs to be mentioned the degrees of freedom and p value in ANOVA.

The fig legends are not clear in most of the figures, especially, Fig 4, 5, 7 that needs to be more clear.

Author Response

Considering the word version of the number of inconsistent lines, we will convert WORD documents to PDF documents, where the number of lines referred to are PDF documents marked by the number of lines.

Point 1: In section 2.6. Data Analysis: I do not know, where the authors used the tukey's test since the they mentioned the Duncan’s test (2.6).

Response 1: Sorry, that is our mistake. We have corrected it. In line 179, significant difference analysis was performed using the Student’s t-test,.

Point 2: In result section 3. Needs to be mentioned the degrees of freedom and p value in ANOVA.

Response 2: Thanks for pointing it to us. We used the Student’s t-test instead of the one-way ANOVA for the analysis of significant differences in the quantitative analysis of TMT proteomics, and the corresponding p-values can be found in Table 4 of the Supplementary Materials.

Point 3: The fig legends are not clear in most of the figures, especially, Fig 4, 5, 7 that needs to be more clear.

Response 3: Thanks, we have revised all of the figure legends in the manuscript to make them clearer.  

Figure 1 legend can be found in Line 260-270.

Figure 2 legend can be found in Line 287-292.

Figure 3 legend can be found in Line 355-357.

Figure 4 legend can be found in Line 367-369.

Figure 5 legend can be found in Line 388-399. 

Figure 6 legend can be found in Line 416-422.

Reviewer 2 Report

Proteome analysis of male accessory gland secretions in the diamondback moth, Plutella xylostella (Lepidoptera: Plutellidae)

The authors study quantitative proteomics between the pre- and post-mated male accessory glands to identify the differences in the proteomic profiles governing the sexual reproduction in this species and develop an ACP database for this insect. To correlate the secretory proteins, authors sequenced the copulatory bursas of the females to identify the true secretory proteins of the male accessory glands. The objective of the study was to study the reproduction biology of these moths to develop novel control methods due to the increased interception of insecticides.

The experiments to address the research question are well executed and the experimental design is well planned. The efforts required to collect the mated individuals is appreciated.

Although, the study fills the gap of knowledge in the reproductive biology of DBM, the study is poorly represented in the current version of the article. The manuscript struggles with clarity of results, figures, and discussion. The statistical approach used to identify DAPs does not use a correction methods for multiple comparisons which increases the false discovery rate. In addition, the comparisons done to virgin MAGs showed only 3 abundant proteins and several less abundant proteins in mated-MAGs. However, the discussion talks about only the functions of the ACPs that are predicted in this study, but not the DAPs. Discussion was heavily weighted on the ACPs and the rationale of performing the differential studies is not clear. 

The manuscript cannot be accepted in the current form. Significant improvement of the results and discussion is required before the manuscript can be published in Insects. Figures and tables have poor resolution and annotation. It is strongly suggested that the authors address these comments before submitting the article.

Please see the PDF with specific comments

Author Response

Response to Reviewer 2 Comments

Considering the word version of the number of inconsistent lines, we will convert WORD

documents to PDF documents, where the number of lines referred to are PDF documents

marked by the number of lines.

Point 1: Italicize the scientific names of the insect species.

Response 1: Thanks, we have italicized all the scientific names of the insect species in the

manuscript, including title, references section. Meanwhile, we have re-added line numbers

for improving the review efforts. Some sentence shifts that may have been caused by the

processing of the original manuscript.

Point 2: Line 23. “are” instead of “were”

Response 2: Thanks for point out this grammar error, we have revised it.

Point 3: Line 23: “seminal fluid” instead of “sperms”

Response 3: We still think the term “sperms” instead of “seminal fluid” should be used. The

male accessory gland proteins (ACPs) account for the majority of Seminal fluid proteins (SFPs)

in most insect species, also referred to as seminal fluid proteins (SFPs). During mating, ACPs

are transferred along with sperms to females. So “sperms” is more appropriate.

Point 4: Line 28: It is not clear how the post-mating state would be helpful for management

practices. Please clarify

Response 4: Thanks, we have deleted that unclear sentence. We’ve revised it to “Mating has

a profound impact on the female's behavior and physiology in this species. It is still unclear

what ACPs are in this species.”

Point 5: Line 36: “regulators” instead of “physiological”

Response 5: Thanks, we have corrected it.

Point 6: Line 73: delete“and analyzation”

Response 6: Thanks, we have deleted it.

Point 7: Line 79: “identified 90 ACPs” instead of “90 ACPs were identified”Response 7: Thanks, we have revised it.

Point 8: Line 107: “under laboratory conditions” in cages in growth chambers? Please

provide specific details.

Response 8: Thanks, we have revised it. Please refer to Line 107-108. We have revised it to

“ in cages with a length, width and height of 50cm×50cm×50cm of the artificial climate

chamber” instead of “under laboratory conditions”.

Point 9: Lines 118: “scotophase” Please include the duration for which they were left at this

phase

Response 9: Thanks, we have revised it. We revised it to “During scotophase” in Line 115.

Point 10: Line 118: “for more than 30 minutes” What was the method employed for timing

the copulation? How was this kept consistent across multiple samples? Were all the 200 per

sex dissections performed on the same day? Please clarify these details.

Response 10: Thanks, we have revised it. We have revised it to “Copulation was examined

every 30 minutes with the faint red light.” in Line 118. Meanwhile, we have added the

description of the method to reduce the error in Line 123-126.

Point 11: Line 132: “SDT” Please explain the abbreviation

Response 11: Thanks, we have revised it. We have revised it to “SDT (4% SDS, 100 mM DTT,

100 mM Tris-HCl)” instead of “SDT”

Point 12: Line 134: “boiling water” Temperature?

Response 12: Thanks, we have revised it. We have revised it to “100 ℃ boiling water”

instead of “boiling water”.

Point 13: Line 172: “Transdecoder” Was the longest ORF taken as the putatively coding

proteins and also, was the ORF annotated to increase the confidence of identification?

Response 13: Thanks, we have revised it. We have added the description of the putatively

coding proteins and increase the confidence of identification in Line 173-177.

Point 14: Line 179: “statistical analysis was conducted using one-way ANOVA, followed by

the Tukey test (P < 0.05)” Was there a multiple comparison correction performed to control

the false discovery rate?Response 14: Sorry, it’s our mistake. We used the Student’s t-test instead of the one-way

ANOVA for the analysis of significant differences. We have revised it to “statistical analysis

was conducted using Student’s t-test (P < 0.05)”.

Point 15: Line 192: “In addition, the DAPs that do not satisfy the above two criteria were

considered as uncertain ACPs” Since the proteins represent ~200 insects, these proteins could

be truly represent ACPs

Response 15: We consider the results to be reliable and these proteins could represent ACPs.

Although at least 200 adult males were dissected for each type of sample, three biological

replicates were prepared for each type of sample.

Point 16: Lines 216: “H. Sapien” Please provide the rationale for using H .sapien SFP

Response 16: Thanks, we have revised it. We have added the rationale for using H .sapien

SFP in Line 216-217.

Point 17: Line 227: “utilizing the ID number” Please provide the ID number here and also for

the rest of the databases

Response 17: The ID number and amino acid sequence are provided in table 1 of the

Supplemental file 1. Meanwhile, we have added “The ID numbers of SFPs for these species

and they can also be obtained in the supplementary material (Additional file 1: Table S1).” in

Line 227-228.

Point 18: Line 248: “Three replicates were prepared for each tissue” Please provide the kits or

brief protocols used for total RNA isolation and cDNA synthesis

Response 18: Thanks, we have added the relevant description in line 248-251: “The total

RNA was isolated using RNAiso Plus (Takara, Dalian, China) based on the instructions

provided by the manufacturer. The reverse transcription was carried out by PrimeScript® RT

Reagent Kit with gDNA Eraser (Takara, Dalian, China) in a reaction mixture of 10 µl with 900

ng of total RNA.”

Point 19: Line 260: Figure 1. Please provide better quality images

Response 19: Sorry, we have increased the resolution of all figures in the manuscript from

300dpi to 600dpi in order to make it clearer.

Point 20: Line 273: “transcripts/proteins” This is confusing. Please clarify if these are transcripts

or proteins

Response 20: Sorry, we have revised it to “proteins” instead of “transcripts/proteins”.Point 21: Line 275: “Among the ANOVA-detected statistically significant transcripts/proteins”

Response 21: thanks, we have revised it to “ Among the Student’s t-test detected statistically

significant proteins”.

Point 22: Line 279: “We selected the longest transcript to represent the unigene when it can be

assembled into different transcript lengths.”

Response 22: we have revised it to We selected the longest CDS to represent the protein

when it contains CDS of different lengths” in Line 279.

Point 23: Line 282: “The farther away from zero that point is, the more significant the

difference. ” Please move this to the figure legend or delete

Response 23: Thanks. We have moved it to the figure 2 legend in Line 291.

Point 24: Line 287: “Figure 2” Please provide better quality figure. Include the description of MAG

in the legend. Explain in the legend what the horizontal and vertical lines stand for.

Response 24: Thanks for your suggestion. We have increased the resolution of all the figure in

the manuscript from 300dpi to 600dpi in order to make it clearer. Also for clarity of

expression, we replaced Figure 2 in Line 286.

Point 25: “Figure 3” Please delete Fig 3 to as the figure 3 by itself can be represented in the

text.

Response 25: Thanks. We have deleted it.

Point 26: Line 309: “Table 1 ” PLease correct the formatting issue in the table

Response 26: Thanks. We have revised the Table 1 and its legend.

Point 27: Line 358: “According to” Include under different subheading

Response 27: Thanks. We have added subheading “Functional classification analysis of

ACPs” in Line 358.

Point 28: Line 371: “Ninety-five ” Include a subheading for the section

Response 28: Thanks. We have added subheading “Comparison of the ACPs of P. xylostella

with other insects” in Line 370.

Point 29: Line 390: “ Jones–Taylor–Thornton (JTT) substitution model” This was not

mentioned in the methodsResponse 29: Sorry for that mistake, we have added it at Line 240.

Point 30: Line 415: Y-axis titles are missing

Response 30: Thanks, we have revised Figure 6 in 415.

Point 31: Line 424: “We made comparisons between male accessory gland immediately before

and after mating using TMT quantitative proteomics, and used shotgun proteomic approach

to analyze the mated female’s copulatory bursas, to identify candidate ACPs. In total, 123

ACPs were identified, and 74 out of 123 of ACPs were predicted to have a signal peptide.”

This paragraph should highlight the overall results identified in terms of biological question

addressed in the study that is comparsion of pre and post mating proteome of males

accessory glands

Response 31: Thanks, we have revised it according to this suggestion. Please refer to Line

424-450 and Line 468-473.

Point 32: Line 434 “74 ACPs in P. xylostella were detected”

Response 32: Sorry, we have revised it “74 out of 123 secreted ACPs had a signal peptide in P.

xylostella” instead of “74 ACPs in P. xylostella were detected”.

Round 2

Reviewer 2 Report

Authors have included the majority of the suggestions except for the statistics employed to identify differentially abundant proteins. A student T-test without a false discovery correction leads to identifying false positives. The authors should redo the statistics to account for the false discovery rate. Also, please include which software or program is being used to perform statistics.

It is still not clear the rationale for the differential expression analysis if the results were not discussed in terms of the ACP composition pre and post mating in males. Please include a rationale for the differential comparison to justify the approach.

Line 571 -587 does not support the results and can be deleted from the discussion

Figure 2 in the revised version is confusing. Please edit it for clarity.

Figure 3 is still not clear

The manuscript cannot be accepted for publication without addressing these details.

Author Response

Response to Reviewer 2 Comments

Considering the inconsistent lines caused by the word version. We will convert WORD documents to PDF documents. In the revised vision the number of lines referred to are the number of lines in the PDF documents.

Comment 1: Authors have included the majority of the suggestions except for the statistics employed to identify differentially abundant proteins. A student T-test without a false discovery correction leads to identifying false positives. The authors should redo the statistics to account for the false discovery rate. Also, please include which software or program is being used to perform statistics.

Response 1: Thanks, we previously adjusted the P values using the Benjamini and Hochberg method. The caculation was completed using the p. adjust function in the R program (version 4.0.3 ) (https://www.r-project.org/). The results showed that q-value (FDR-adjusted P value) was less than 0.01, only 2 differential abundance proteins were detected. When q value < 0.05, only 74 differential abundance proteins were identified, of these, 15 proteins were unconfirmed ACPs and 7 proteins were functionally unknown due to they did not contain conserved domains. The excessively strict filtering conditions may result in the loss of much useful information, which is not very suitable for existing proteomics studies. In this study, the q-value of the differentially abundant proteins was a maximum of 0.231. Therefore for proteomics data analysis, Student’s t-test is the most commonly used test in this type of data analysis, which may not account for false discovery rate, and there are many published papers that used this statistic method (Some references are listed below).

References:

  1. Du Y, Wang Y, Xu Q, Zhu J, Lin Y. TMT-based quantitative proteomics analysis reveals the key proteins related with the differentiation process of goat intramuscular adipocytes. BMC Genomics 2021, 22, 417. 
  2. Huang W, Huang P, Yü D, Li C, Huang S, Qi P, Huang S, Keyhani NO, Huang Z. Proteomic Analysis of a hypervirulent mutant of the insect-pathogenic fungus metarhizium anisopliaereveals changes in pathogenicity and terpenoid p Microbiol. Spectr. 2022, 10, e0076022.
  3. Li L, Huang Q, Barbero M, Liu L, Nguyen T, Xu A, Ji L. Proteins and signaling pathways response to dry needling combined with static stretching treatment for chronic myofascial pain in a RAT model: an explorative proteomic sInt. J. Mol. Sci. 2019, 20, 564.
  4. Liu J, Sui Y, Chen H, Liu Y, Liu Y. Proteomic analysis of kiwifruit in response to the postharvest pathogen, Botrytis cinerea. Front.Plant Sci. 2018, 9,158.

Therefore, we have revised it to “For quantitative proteomic analysis of virgin- versus mated-MAGs, statistical analysis was performed according to the method described in these studies [29,30,31,32]. Both fold change value of greater than ± 1.5 and p < 0.05 were used to identify the differentially abundant proteins (DAPs)”.

Comment 2: It is still not clear the rationale for the differential expression analysis if the results were not discussed in terms of the ACP composition pre and post mating in males. Please include a rationale for the differential comparison to justify the approach.

Response 2: Thanks, We have revised it. Due to the proteomics study in this study was based on the change in differential protein abundance before and after mating for the identification of ACPs, so we have revised it to “In insect species, ACPs are synthesized in the accessory gland of males and transferred to females during mating, and they trigger multiple physiological and post-mating behavioral changes”. ”We performed TMT quantitative proteomics to identify male accessory glands produced ACPs in P. xylostella, by comparing changes in the abundance of ACPs before and after mating immediately. In total, 168 ACPs were identified. Among those ACPs, 165 ACPs were significantly down-regulated and 3 ACPs were significantly up-regulated. Most of the ACPs are down-regulated proteins, which also demonstrated that ACPs were transferred to females during mating and the ACPs identification method used in this study was efficient” Please refer to Line 425-452.

Comment 3: Line 571 -587 does not support the results and can be deleted from the discussion

Response 3: Thanks, we have deleted it.

Comment 4: Figure 2 in the revised version is confusing. Please edit it for clarity. Figure 3 is still not clear

Response 4: Sorry, this may be caused during word to PDF processing. We have contacted the assistant editor, Ms. Jessie Jiao, and she will help us to insert sharp high resolution figures into the revised version of manuscript in the appropriate places.